# Voriconazole Cyclodextrin Based Polymeric Nanobeads for Enhanced Solubility and Activity: In Vitro/In Vivo and Molecular Simulation Approach

**DOI:** 10.3390/pharmaceutics15020389

**Published:** 2023-01-24

**Authors:** Mudassir Farooq, Faisal Usman, Mahrukh Naseem, Hanan Y. Aati, Hassan Ahmad, Sirikhwan Manee, Ruqaiya Khalil, Kashif ur Rehman Khan, Muhammad Imran Qureshi, Muhammad Umair

**Affiliations:** 1Department of Pharmaceutics, Faculty of Pharmacy, Bahauddin Zakariya University, Multan 60800, Pakistan; 2Department of Zoology, University of Balochistan, Quetta 08770, Pakistan; 3Department of Pharmacognosy, College of Pharmacy, King Saud University, Riyadh 11495, Saudi Arabia; 4Faculty of Pharmaceutical Sciences, University of Central Punjab, Lahore 54000, Pakistan; 5Traditional Thai Medicine Research and Innovation Center, Faculty of Traditional Thai Medicine, Prince of Songkla University, Songkhla 90110, Thailand; 6Centro De Investigaciones Biomédicas, University of Vigo, 36310 Vigo, Spain; 7Department of Biochemistry, Genetics and Immunology, University of Vigo, 36310 Vigo, Spain; 8Department of Pharmaceutical Chemistry, Faculty of Pharmacy, The Islamia University of Bahawalpur, Bahawalpur 66000, Pakistan; 9Department of Pharmaceutics, Faculty of Pharmacy, The Islamia University of Bahawalpur, Bahawalpur 66000, Pakistan; 10College of Pharmacy, Shenzhen Technology University, Shenzhen 518060, China

**Keywords:** voriconazole, hydroxy propyl β cyclodextrin, fungal infection, nanobeads, free radical polymerization

## Abstract

Hydroxypropyl β-cyclodextrin (HPβCD) based polymeric nanobeads containing voriconazole (VRC) were fabricated by free radical polymerization using *N, N*′-methylene bisacrylamide (MBA) as a cross-linker, 2-acrylamide-2-methylpropane sulfonic acid (AMPS) as monomer and ammonium persulfate (APS) as reaction promoter. Optimized formulation (CDN5) had a particle size of 320 nm with a zeta potential of −35.5 mV and 87% EE. Scanning electron microscopy (SEM) depicted porous and non-spherical shaped beads. No evidence of chemical interaction was evident in FT-IR studies, whereas distinctive high-intensity VRC peaks were found superimposed in XRD. A stable polymeric network formation was evident in DSC studies owing to a lower breakdown in VRC loaded HPβCD in comparison to blank HPβCD. In vitro release studies showed 91 and 92% drug release for optimized formulation at pH 1.2 and 6.8, respectively, with first-order kinetics as the best-fit model and non-Fickian diffusion as the release mechanism. No evidence of toxicity was observed upon oral administration of HPβCD loaded VRC polymeric nanobeads owing to with cellular morphology of vital organs as observed in histopathology. Molecular docking indicates the amalgamation of the compounds highlighting the hydrophobic patching mediated by nanogel formulation. It can be concluded that the development of polymeric nanobeads can be a promising tool to enhance the solubility and efficacy of hydrophobic drugs such as VRC besides decreased toxicity and for effective management of fungal infections.

## 1. Introduction

Poor aqueous solubility, lack of stability, and effective targeting at the desired site are challenging issues for drug design. Currently, 90% of new drug moieties and 40% of the existing therapeutic agents are not soluble in aqueous media, which poses difficulty in attaining desired therapeutic concentration in biological fluids. Such substances demonstrate variable absorption and bioavailability. One of the biggest issues facing modern drug development researchers is the low water solubility of active pharmaceutical ingredients [1].

Voriconazole (VRC) is a synthetic, second-generation triazole antifungal drug that binds with ergosterol, inhibits fungal cytochrome P450 enzyme, and prevents cell membrane formation. It has an excellent spectrum against *Candida, Aspergillus,* and *Fusarium* species. VRC belongs to BCS-II, having low solubility and high permeability [2]. The t_1/2_ of VRC is 2.5–3 h, and its bioavailability is 96% by oral route when taken empty stomach. Its solubility is 0.7 mg/mL in water and 2 mg/mL in methanol.

Nanobeads are a cross-linked 3-dimensional polymeric network that absorbs the maximum quantity of water. They have been in the pharmaceutical field owing to maximum drug loading capacity, enhanced solubility, excellent swelling behavior, biocompatibility, and desirable chemical properties [3]. Nanobeads have stability for a prolonged duration, novel functionality, and sustained release effect by incorporating a high-affinity functional group [4].

Cyclodextrins (CD) are crystalline, non-reducing sugars obtained by the breakdown of starch. CD contains glucopyranose units linked by 1, 4-glycosidic bonds. CD’s outer surface is hydrophilic, while the inner cavity is hydrophobic [5]. CD increases stability and solubility for oral drug delivery. CD also acts as a mucoadhesive agent. CD forms a complex in the aqueous state by incorporating drugs into their cavity. When a complex is formed, drug molecules are in equilibrium with non-complex molecules in the solution. The formation of initial equilibrium is rapid, while final equilibrium takes a longer time to stabilize during complex formation. When the drugs enter the CD cavity, conformational changes occur to gain maximum benefit from the presence of weak Van der Waal forces. The forces behind complexation are hydrogen bonding, Van der Waals, hydrophobic, electrostatic, and charge-transfer interactions. The active drug complexation depends on its molecular size and functional groups [6]. Based on their structure, α-CD forms a complex with lower molecular weight drug, β-CD complexed with aromatic and heterocyclic molecules, and γ-CD retains larger molecules (macrocycles and steroids). The γ-CD contains the largest hydrophobic center, a more significant therapeutic profile, and high solubility than other CDs.

Many hydrophilic derivatives of CD are being used, including sulfobutyl, methyl, and hydroxypropyl derivatives. HPβCD is the most suitable and harmless for oral, external, and parenteral drug administration. Hydroxypropyl derivatives have low hygroscopicity [7]. Recent studies indicate that the inclusion complex formation can be increased by utilizing hydrophilic monomers such as hydroxypropyl methylcellulose, polyethylene glycol, and polyacrylic amide. The solubility and bioavailability of drugs can be enhanced by polymerization of CD with monomer, i.e., 2-acrylamide-2-methylpropane sulfonic acid (AMPS), methacrylic acid (MAA) by hydrogel formation. Swelling of complex formation is pH-dependent, pH-independent, and ionic-responsive behavior [8,9].

AMPS has hydrophilic nature and serves as the best carrier for pH-independent swelling. Moreover, AMPS is currently used in foam stabilizers, biomedical engineering, carriers in muscle actuators, and drug delivery. Ammonium persulfate (APS) is a water-soluble organic compound having a non-explosive and highly economical compound consisting of one peroxydisulfate anion and two ammonium cations. It is used as an initiator in free radical polymerization and *N, N*′-methylene bis acrylamide (MBA) as a cross-linker [10]. In the current study, we developed HPβCD based polymeric nanobeads for increasing the solubility of VRC to attain better therapeutic effects with reduced toxicity.

## 2. Materials and Methods

### 2.1. Materials

VRC was gifted from Ferozsons Laboratories (Pvt.) Ltd. Nowsherea, KPK, Pakistan. Ethanol, HPβCD, AMPS, APS, and MBA were purchased from Sigma Aldrich, Spruce St Saint Louis, MO, USA. Distilled water was obtained from AASS pharmaceutical Multan, Punjab, Pakistan. All the chemicals used were of analytical grade and used without further modification.

### 2.2. VRC-CD Inclusion Complex Preparation

HPβCD based polymeric nanobeads were prepared by free radical polymerization using MBA as a cross-linking agent and APS as a reaction promoter. Aqueous solutions of HPβCD and AMPS were prepared in 10 mL of distilled water. APS and MBA were accurately weighed and dissolved in distilled water separately. AMPS solution was added dropwise in HPβCD solution during stirring. Afterward, APS was added as an initiator in the previous mixed solution at 600 rpm. Finally, MBA was added at 45 °C. After uniform mixing, the whole solution was sonicated (FSF-020S), vortexed (ZX3, VELP Scientifica, Usmate Velate, Italy), and purged using nitrogen gas to ensure appropriate mixing and removal of entrapped air. Finally, the mixed aqueous solution was shifted to Petri dishes. Petri dishes were wrapped with aluminum foil and kept in a hot air oven at 45 °C for 4 h followed by 65 °C for 24 h to initiate the polymerization reaction. After drying, a hard transparent solid mass was removed from the Petri dishes and washed with distilled water to remove unreacted moieties until constant pH was attained. Finally, fabricated formulations were passed through sieve No. 80 to obtain uniform beads. These beads were lyophilized and stored in an air-tight container for further experiments. The proposed chemical reaction of complex formation is represented in Appendix A Various concentrations of HPβCD, MBA, and AMPS used for formulations are given in Table 1 [11,12].

#### Drug Loading of CD-Based Polymeric Nanobeads

VRC was loaded in HPβCD-based polymeric nanobeads by the diffusion-assisted swelling method. Drug solutions of different concentrations were prepared in 50% ethanol and water and poured into previously weighed hydrogel nanobeads in the Petri dish. After 12 h, the nanobeads were filtered and washed with water to remove VRC content from the surface. Finally, nanobeads were dried in a hot air oven at 45 °C [13].

### 2.3. Characterization of Cyclodextrin-Based Polymeric Nanobeads

#### 2.3.1. % Entrapment Efficiency and Product Yield

VRC HPβCD based polymeric nanobeads were weighed equivalent to 10 mg of pure VRC and put in 50% ethanol–water solution that was previously used for drug loading. A total of 40 mL of ethanol–water solution was employed for VRC extraction and refluxed for 1 h to facilitate the release of VRC from nanobeads. Samples were withdrawn, filtered, and VRC content was evaluated by UV-Visible Spectrophotometer (UV-1900i, Shimadzu, Nakagyo-ku, Kyoto, Japan) at 245 nm using Equation (1) [14]. The product yield of VRC CD-based polymeric nanobeads was calculated using Equation (2).
(1)$$\%\;\mathrm{Entrapment}\;\mathrm{efficiency}=\frac{\mathrm{The}\;\mathrm{absorbance}\;\mathrm{of}\;\mathrm{nanobeads}\;\mathrm{sample}}{\mathrm{Absorbance}\;\mathrm{of}\;\mathrm{pure}\;\mathrm{VRC}}\times \;100\;$$
Product yield = 100% − weight loss (2)
(3)$$\%\;\mathrm{Weight}\;\mathrm{loss}=\frac{W_{0-}W_{i}}{W_{0}}\times 100$$

*W_o_* is the total amount of drug loaded, and *Wi* indicates the drug present in a sample.

#### 2.3.2. Solubility Enhancement

Solubilization enhancement of HPβCD-based polymeric nanobeads was evaluated using RO water and HCL and Phosphate buffer solutions of pH 1.2 and 6.8. All suspensions were kept for 24 h on a magnetic stirrer at 26 °C. At that point, suspensions were centrifuged and sifted through the Whatman filter (0.45 μm). The resultant samples were analyzed by UV-Visible Spectrophotometer at 245 nm [15].

#### 2.3.3. Swelling Behaviour in Water, pH 6.8 and 7.4 Buffers

The dried polymeric nanobeads were weighed and stirred for 24 h at pH 1.2 and pH 6.8 at room temperature. Afterward, the resultant wet mass achieved a constant weight upon maximum swelling, and equilibrium swelling was measured using Equation (4).
(4)$$\%\;\mathrm{Swelling}\;\mathrm{ratio}\;𝑞=\frac{W_{s}}{W_{d}}\;\times \;100\;$$

𝑊𝑠 represents swollen nanobeads’ weight, and $W_{d}$ indicates dry nanobeads’ weight [16].

#### 2.3.4. Fourier Transformed Infrared Spectroscopy

The molecular interaction of VRC with HPβCD, AMPS, APS, and MBA was evaluated by infrared spectroscopy FTIR spectra were recorded on an IR spectrophotometer (PerkinElmer Lambda 7600S, Glen Waverley VIC, Australia) in the range 4000 ${\mathrm{cm}}^{-1}$ to 500 ${\mathrm{cm}}^{-1}$ [17].

#### 2.3.5. Scanning Electron Microscopy (SEM)

VRC HPβCD-based polymeric nanobeads’ surface morphology was studied using a scanning electron microscope (GeminiSEM 560, ZEISS, Jena, Germany). The samples were placed on an aluminum pan coated with gold.

#### 2.3.6. Particle’s Size and Zeta Potential

Zeta potential and particle size were measured by a zeta sizer analyzer (Nano ZS-90, Malvern, UK) to assess the stability of the formulation. A transparent, disposable zeta cell was filled with an aqueous suspension of nanobeads in Milli-Q water [18].

#### 2.3.7. Differential Scanning Calorimetry

The DSC of VRC, HPβCD, blank, and loaded HPβCD-based polymeric nanobeads were carried out to quantify the heat of fusion. Samples were encased in an aluminum plate after trituration. Equipment (SDT Q600 V8.2, Shimadzu, Kyoto, Japan) was operated at 10 °C/min under a stream of nitrogen from 0–400 °C.

#### 2.3.8. Thermal Analysis

TGA thermal analysis was performed on instrument (SDT Q600 V8.2, Shimadzu) to measure physio-chemical changes of VRC, HPβCD, and HPβCD based polymeric nanobeads under controlled conditions [19].

#### 2.3.9. Powder X-ray Diffraction (XRD)

Using an X-ray diffractometer (JDX-3523, JEOL, Tokyo, Japan), a powder X-ray Diffraction (PXRD) study of pure drug, HPβCD, loaded and unloaded HPβCD based polymeric nanobeads were performed. All the powdered materials were packed securely into an aluminum cell and subjected to Cuka monochromatic radiations of wavelength 1.54056 A°. Samples were examined between 5° and 60° using 2θ at a rate of 3°/min.

### 2.4. In Vitro Dissolution Studies

VRC-loaded HPβCD polymeric nanobeads were analyzed for release behavior in pH 1.2 and 6.8 buffer solutions using USP Dissolution Apparatus II at 37 ± 0.5 °C. A sample of VRC polymeric nanobeads was weighed and packed in a capsule shell. Similarly, pure VRC was weighed and sealed in a capsule. Both samples were separately run in dissolution media at 50 rpm for 3 h in acidic media and 3 h in pH 6.8 phosphate buffer. Samples were withdrawn at equal time intervals and analyzed at 245 nm [20].

### 2.5. Drug Release Kinetics

In order to determine the release mechanism of VRC polymeric nanobeads, various Pharmacokinetic models were applied to in vitro release data, as given in Equations (5)–(8).
(5)$$\mathrm{Zero}\text{-}\mathrm{order}\;\mathrm{kinetics},\;\mathrm{Ft}=k_{0}t\;$$
(6)$$\mathrm{First}\text{-}\mathrm{order}\;\mathrm{kinetics},\;\ln\;(1-F)=-K_{1\;}t$$
(7)$$\mathrm{Higuchi}\;\mathrm{model}\;F=K_{2\;}t{\;}_{½}\;$$
(8)$$\mathrm{Korsmeyer}–\mathrm{Peppas},\;\frac{\mathrm{Mt}}{M}=K_{3}\mathrm{tn}\;$$

Ft and F are a fraction of released drugs and released a fraction of drugs in time t. K_0,_ $K_{1\;}$ K_2_ and K_3_ represent zero-order, first-order release rate, Higuchi constant, and constant incorporate. M and M_t_ refer to the water mass absorbed at equilibrium and the mass of water absorbed in time (t). n represents the release exponent [21].

### 2.6. Acute Oral Toxicity Studies

#### 2.6.1. Animal Housing

Fifteen (15) healthy adult male rabbits were taken from Animal House, Faculty of Pharmacy, Bahauddin Zakariya University, Multan, Punjab, Pakistan. Before beginning the experiment, rabbits were given one week of acclimation. Three groups, with each group having five (5) rabbits, were randomly assigned a well-ventilated wooden cage. All animals had access to food and water without restriction. The quarantine area’s temperature was maintained at 25 ± 2 °C using a 12 h light/dark cycle.

#### 2.6.2. Sampling

Intragastrically injected normal saline was given to Group A as a control. Both the unloaded and the loaded HPβCD based polymeric nanobeads (5 mg/kg of body weight) were delivered intragastrically to groups B and C. For 12 h before administering the dosage, animals were put on a feed fast. The OECD’s criteria were followed in conducting the toxicology investigation.

#### 2.6.3. Clinical Manifestations

Individually, all animals were evaluated for general conditions, including alertness and grooming, convulsions and hyperactivity, lacrimation, salivation, urine, touch reaction, pain response, writhing reflex, corneal reflex, grasping strength, and righting reflex. Food and water consumed by each group were also determined.

#### 2.6.4. Blood Analysis

Exsanguination and euthanasia were performed on all rabbits on the fifteenth day. Blood was drawn from the supraorbital vein into an EDTA-K2 blood collection tube for hematological testing. The results of the blood tests were recorded using the hematological analyzer. The blood sample was centrifuged at 3000 rpm for 10 min, and the biochemical characteristics of the serum were evaluated using a clinical chemistry analyzer. Rabbit necropsies were performed. Analytical balances were used to weigh the animal’s vital organs. Vital organs were maintained in a 10% formaldehyde solution for 72 h before histological analysis. Afterward, the tissue was cut into pieces and dehydrated using ethanol. Slides were prepared and stained using Hematoxylin and Eosin (H and E) and observed using an optical microscope.

### 2.7. Molecular Docking

The structure of VRC has been retrieved from the Drug bank and charged and minimized (force field: MMFF94x) using the MOE software suite. [Molecular Operating Environment (MOE), 2018.10; Chemical Computing Group ULC, 1010 Sherbooke St. West, Suite #910, Montreal, QC, Canada, H3A 2R7, 2018]. In MOE, the chemical structures of AMPS and HPβCD were designed using the builder module and were further charged and minimized using the minimizing module. Molecular modeling was performed following the stoichiometric calculations of CDN5 using the default rigid receptor protocol in MOE. All the calculations were performed at a constant temperature of 310 K in a vacuum (pH = 7.2). The resulting poses were analyzed visually. All the graphics were rendered using the NGL viewer [22].

### 2.8. Antifungal Activity

*Candida albicans*, *Aspergillus flavus*, and *Aspergillus fumigatus* were used for antifungal activity. The hole-punch method was used to investigate the antifungal properties of polymeric nanobeads. VRC and HPβCD based polymeric nanobeads were dissolved in polyethylene glycol. A diluted stock solution in RPMI 1640 medium 0.165 M morpholine propane sulfonic acid was added to 7.0 pH. After mixing, 100 mL of 0.15 mg/mL VRC and HPβCD based formulation solution in phosphate buffer saline was taken. Yeast inocula were grown for 24 h at 35 °C on sabouraud dextrose agar plates to concentrations of 0.6 g–2.6 g. For 1–2 days, Yeast inocula were added to microdilution plates containing 100 mL of drug solution. The MIC and MFC were determined [23,24].

### 2.9. Statistical Analysis

In order to assess the degree of significance, the data were statistically evaluated. Unless otherwise specified, data are provided as the mean and standard deviation (SD) of at least five samples. Analysis of variance (ANOVA) was used to examine the data, and then additional appropriate statistical parameters as needed. The threshold for statistical significance was set at *p* < 0.05 [25].

## 3. Results

### 3.1. % Entrapment Efficiency and Product Yield

The maximum quantity of VRC was observed for CDN5, i.e., 87%. Product yield, whereas % EE results were in the range of 82–94% and 72–87%, as shown in Table 2. Polymer concentrations affect the % EE; % EE increases with increasing polymer concentration [26]. The formulation CDN5 showed maximum % EE due to the repulsion of sulfonic acid groups [27].

### 3.2. Solubility Improvement

The solubility of VRC HPβCD polymeric nanobeads was potentiated at pH 1.2, pH 6.8, and in deionized water. The solubility of nanobeads formulation was more in aqueous media as compared to pH 1.2 and pH 6.8 buffer, as shown in Figure 1.

### 3.3. Swelling Behaviour in Water, pH 6.8 and 7.4 Buffers

Release of VRC from HPβCD polymeric nanobeads was based on swelling of nanobeads; Maximum release of VRC was observed on maximum swelling of nanobeads. Fabricated polymeric nanobeads had higher swelling at pH 6.8. The % swelling ratio was between 14.2% and 91.2% at pH 1.2, while at pH 6.8 swelling ratio was between 16.3% and 96.9%, as given in Figure 2.

### 3.4. Fourier Transformed Infrared Spectroscopy

Infrared spectra were analyzed by comparing VRC, HPβCD, Blank, and loaded HPβCD based polymeric nanobeads. VRC showed distinct absorption peaks with substantial aromatic bending at 860 cm^−1^, 612 cm^−1^, C-N stretching at 1274 cm^−1^, and C-C at 1584 cm^−1^. C=C at 1452 cm^−1^, and C-O stretching at 1124 cm^−1^. At 3191 cm^−1^, VRC showed weak OH stretching. In pure HPβCD, the strong, wide absorption band of free OH stretching vibration at 3356 cm^−1^ was observed. The peak shows the vibration of CH_3_ at 2922 cm^−1^. Other notable absorption bands may be seen at 1641 cm^−1^, 1155 cm^−1^, and 1017 cm^−1^, which represent H-OH bonding, C-O, and C-O-C stretching vibrations, respectively. The visible bands of HPβCD were displaced in the IR spectra of blank polymeric nanobeads. Characteristic absorption bands 1155 cm^−1^ demonstrated that blank HPβCD based polymeric nanobeads contained HPβCD as the backbone with a sulfate group as a side chain. The ester sulfate stretching of AMPS is responsible for this peak. The N-H stretching bands overlapped with the OH stretching band of the polymeric nanobeads. At 1548 cm^−1^, a new peak of the ether group connected to the carbonyl group was seen, indicating the formation of a drug-polymer complex. Since the new polymer was generated through cross-linking of monomer and polymer, alteration of some of the drug’s distinctive bands shows that only that portion of the molecule has been complexed with HPβCD.

On the other hand, Bands relating to the non-complex component remain unchanged. The sharp absorption band at 1641 cm^−1^ has been shifted to 1616 cm^−1^. Absorption peaks at 1274 cm^−1^, 1124 cm^−1^, and 1143 cm^−1^ vanished or were submerged by peaks of loaded polymeric nanobeads, as shown in Figure 3. VRC was loaded into the hydrophobic cavity of HPβCD, which was consistent with the XRD data.

### 3.5. Scanning Electron Microscopy (SEM)

VRC HPβCD based polymeric nanobeads were very porous, compact, and spongy in appearance. Because of the porous nature, rough surface, and tiny size, water was absorbed instantly, resulting in immediate swelling and the release of VRC. Figure 4a,b depicts the irregular and rough surface having filled pores of a drug, while Figure 4c shows blank pores, indicating the absence of the drug. The prepared nanobeads had non-spherical shapes and varied sizes, resulting in a wide surface area, fast swelling, and improved solubility.

### 3.6. Particle Size and Zeta Potential

The particle size of VRC-HPβCD-based polymeric nanobeads was in the range of 220–436 nm. The optimized formulation CDN5 was 220 nm in size with PDI 0.23. The zeta potential of the optimized formulation was −35.1 mV indicating a stable formulation with low aggregation, as illustrated in Table 3 and particles size are graphically shown in Appendix A.

### 3.7. Thermal Analysis

The Pure VRC has a melting endotherm of 133 °C, and the HPβCD curve showed a minor endothermic peak at 112 °C with an enthalpy change of 77.26 J/g, as shown in Figure 5a,b. The lack of an endothermic peak can be seen in the DSC curve of blank HPβCD polymeric nanobeads in Figure 5c due to the absence of the drug. These changes in thermal properties followed chemical cross-linking inside blank polymeric nanobeads. It is significant that, as a whole, decomposition was far less in blank HPβCD polymeric nanobeads as compared to VRC loaded HPβCD polymeric nanobeads. This is because HPβCD experienced morphological changes during chemical cross-linking, which altered its structure and properties. When the TGA of HPβCD based polymeric nanobeads were compared to pure HPβCD, the overall breakdown was lower, suggesting the creation of a more stable polymeric network, as shown in Figure 6.

### 3.8. Powder X-ray Diffraction

The solid-state properties of newly produced blank and loaded HPβCD based polymeric nanobeads were investigated. Because the drug was in a solid state, its solubility and dissolution were greatly affected [28]. High-intensity peaks were not seen in the diffractograms of HPβCD or HPβCD based polymeric nanobeads, as illustrated in Figure 7. Using diffractograms, it was clear that they were amorphous. It was discovered that the diffraction patterns of HPβCD -based polymeric nanobeads were different from the pure HPβCD diffraction patterns. Additional amorphous sites were formed when hydrophilic monomers were added to a raw polymer. This led to the assumption that efficient polymerization was responsible for nanobeads’ amorphous nature. A diffractogram of VRC showed high-intensity peaks at several diffraction angles, indicating that the material is crystallized.

### 3.9. In Vitro Dissolution Studies

Dissolution studies showed variation in the release of VRC from polymeric nanobeads compared to pure VRC. VRC was ionized in basic media due to its weak acidic nature. The pure VRC was released up to 52%, while in CDN5, 91% of VRC was released in 3 h as revealed in Figure 8. There was a bit of difference in the release pattern at pH 6.8 owing to pH-independent swelling, as manifested in Figure 9. Drug release from polymeric nanobeads was rapid but erratic based on the monomer, polymer, and pH of the media. The abrupt and maximum release was seen within 3 h (CDN5), ranging from 11.8–91%. Maximum and rapid release profiles were observed by increasing AMPS concentration (3–7%) in CDN5.

### 3.10. Drug Release Kinetics

In vitro findings were analyzed using drug release kinetic models zero order, first order, Higuchi, and Korsemeyer–Peppas to determine the most suitable model and mechanism for VRC released from polymeric nanobeads and pure VRC at pH 1.2 and 6.8. At pH 1.2 (0.99) and pH 6.8, VRC HPβCD based polymeric nanobeads were seen first order as the best-fit model indicated by R^2^ values (0.99). Furthermore, at pH 1.2, the value of “n” was 0.67, and at pH 6.8, it was 0.68, indicating non-Fickian diffusion, as depicted in Table 4. The small particle size, increased wettability, large porous surface area, and pH-independent swelling behavior were all affected the dissolution rate [29].

### 3.11. Acute Oral Toxicity Research

The HPβCD based polymeric gel was demonstrated to have no tissue-irritating effects following oral administration with enhanced solubility of the hydrophobic drug. The main concerns about the toxicity of HPβCD based polymeric nanobeads are persistent unreacted monomer and initiator. As a result, it is critical to evaluate the biocompatibility and toxicity of the delivery mechanism. Standard in vivo oral acute toxicity experiments were used to assess the safety of both blank HPβCD based polymeric nanobeads and loaded HPβCD based polymeric nanobeads. The findings of this work will be helpful in future sub-chronic and chronic gastrointestinal toxicity investigations of HPβCD polymeric nanobeads, which may lead to its use as a drug with a better safety profile.

#### 3.11.1. Clinical Signs and Symptoms

Throughout the length of the trial, conventional toxicity symptoms in Groups A, B, and C were monitored. There were no toxic symptoms linked with the administration of both Blank and loaded HPβCD based polymeric nanobeads. There was no variation in the behavior of rabbits in either group. The administered and control groups consumed equal quantities of food and water. Rabbit weight increased steadily in all groups during the trials, and no difference between the control and treatment groups was observed. Furthermore, no animal died in any of the groups during the trial.

#### 3.11.2. Hematological Examination

Blood samples were obtained from the abdominal aorta at the end of the trial and utilized to assess hematological indicators, as reported in the acute toxicity. Table 5 lists the hematology indicators that were tested using whole blood. All parameters in both administered groups were within the normal range.

#### 3.11.3. Biochemical Examination

Serum from different blood samples was tested, and biochemistry indicators were measured. Alanine aminotransferases (ALT) and aspartate aminotransferase (AST) levels are greatly elevated in a toxic environment due to tissue damage in the liver and kidney. Table 5 showed that all values were within the normal range, indicating that there was no toxicity in the blood as well as hepatic tissues. As a result, it was established that the vital organs of all rabbits in the control and administered groups were normal.

#### 3.11.4. Histopathological Examination

There were no macroscopic alterations found during a gross assessment of critical organs. Table 6 displayed the weight of important organs from necropsied rabbits of all groups. The weight of vital organs did not change significantly between the control and administered groups of necropsied rabbits. Histopathological examinations were performed to observe any evidence of toxicity in the supplied dose form at the cellular level. Figure 10a showed that there was no necrosis, inflammatory infiltration, localized degeneration, or fibrous tissue hyperplasia in any necropsied rabbit’s cardiac tissues. In either group, no clinical abnormalities such as chronic inflammatory infiltration, hyperemia, or vacuolar atrophy of liver cells were identified.

Pathological abnormalities such as bleeding, interstitial infiltration, and stiffness of walls of blood vessels were not found in the lungs of any of the rabbits in the control or treated groups. Figure 10e showed that the kidneys of any necropsied rabbit had no nephropathy, dilated globules, atrophied glomerulus, or renal, tubular, or interstitial calcification. Histopathological study of the stomachs of all necropsied rabbits revealed no signs of gastric toxicity, including localized bleeding of the gastric mucosa and cystic dilatation of the gastric glands. There was no evidence of localized inflammation or duodenal in any of the rabbits from the control or treated groups, indicating no structural alterations in organs that could be attributed to the administration of blank and loaded HPΒCD based polymeric nanobeads. The microscopy of essential organs was comparable with the results of hematological and biochemical indicators (shown in Appendix A) assigned to normally functioning organs [30].

### 3.12. Molecular Modelling

Molecular modeling studies were carried out to establish the binding mode of the ligand and to gain insight into the molecular forces behind the microgel formulation. Figure 11 shows the plausible binding mode of the VRC with the HPβCD and AMPS molecules. As obvious from Figure 11, the nanobeads formation agent has wrapped the VRC, masking the electronegative chloride ions and increasing the hydrophobicity of the complex. The observation is consistent, which explains the sharp changes in the angles observed in XRD. Further, the aliphatic chains of the AMPS and HPβCD are involved in hydrophobic interaction with the azoles and difluoro-benzene rings. The hydrophobic interaction decreases the water-octanol solubility of the complex, which is reported to increase bioactivity by improving bioavailability. Figure 12 indicates the amalgamation of the compounds highlighting the hydrophobic patching mediated by the nanobeads formulation.

### 3.13. Antifungal Activity

A significant increase in the antifungal activity of HPβCD based polymeric nanobeads was observed as compared to pure VRC. The MIC and MFC of CDN5 against *Aspergillus fumigatus* were 0.82 µg/mL and 3.4 µg/mL as compared to pure VRC having MIC and MFC were 1.1 µg/mL and 4 µg/mL, respectively. Similarly, In the case of *Aspergillus flavus* and *Candida albicans*, NIC and MFC were also significantly decreased as compared to pure VRC. The results are graphically presented in Figure 13.

## 4. Discussion

The formulation CDN5 showed maximum % EE owing to the repulsion of sulfonic acid groups surrounded by polymeric complex, and maximum swelling of nanobeads was observed [31]. H Shoukat et al., 2020 also reported similar results. As a cross-linker, MBA helps create compact, dense networks with smaller mesh sizes, which lowers the % EE of VRC [32].

The solubility was improved in formulations CDN5 with increasing AMPS content. Additionally, a rise in AMPS concentration in CDN6 indicated a decline in solubility. This drop occurred because a dense network created by an increase in AMPS concentration prevented the full quantity of the drug from being integrated [33]. The formulation with CDN5 improved maximum solubility. This greatest increase in solubility may be correlated to AMPS’ pH-independent swelling. Maximum swelling was observed in formulation CDN5, as stated in a study on swelling. This maximal swelling caused the polymeric structure to expand to its maximum size and the drug to dissolve completely, resulting in increased solubility [34]. The drug’s solubility profile increased up to 8 times at pH 1.2, 7 times at pH 6.8, and 6.5 times in deionized water in CDN5. The drug’s ability to become more soluble was adversely reduced by higher cross-linker (MBA) concentrations. A dense and compact system was produced by raising the cross-linker concentration. Drug absorption was low, and instead, the drug’s solubility was only marginally improved due to the dense and compact system’s inability to swell correctly and thoroughly [35]. Due to the presence of a significant number of hydroxyl groups in polymer chains that form hydrogen bonds with water, the swelling ratio increased as CD concentrations increased. When AMPS units are introduced to nanobeads, their hydrophilicity and water absorption increase because AMPS is a hydrophilic monomer having strongly ionizable sulfonate groups that completely dissolve at all pH ranges. By increasing MBA concentration, the swelling was significantly minimized. This is so because cross-linkers encourage the physical tangling of polymeric chains, creating a denser, more compact network that is hydrophobic and prevents water from penetrating it. Similar findings on the influence of MBA on the swelling index were also found by Xia et al. [36,37].

The FTIR results showed the formation of polymeric gel that was consistent with XRD data. The prepared nanobeads had non-spherical shapes and varied sizes, resulting in a wide surface area, fast swelling, and improved solubility. Increased surface area, the ability to include medications that are poorly soluble in water into formulations, and quick water absorption are benefits associated with nano-sized particles [38]. The zeta potential of the optimized formulation was −35.1 mV indicating a stable formulation with less chance of aggregation. The PDI value of 0.23 indicated the monodispersity of polymeric nanobeads with better stability [39]. The lack of an endothermic peak can be seen in the DSC curve of blank VRC HPβCD polymeric nanobeads. These changes in thermal properties followed chemical cross-linking inside blank nanobeads. It is significant that, as a whole, decomposition was far less in blank HPβCD hybrid polymeric nanobeads as compared to HPβCD, indicating the formation of a more stable polymeric network. This is because HPβCD experienced morphological changes during chemical cross-linking, which altered its structure and properties. When the TGA of HPβCD based polymeric nanobeads were compared to pure HPβCD, the overall breakdown was lower, suggesting the creation of a more stable polymeric network. When compared to pure HPβCD, the TGA thermogram showed that the commencement of breakdown was initiated first in blank HPβCD based polymeric nanobeads.

The dissolution rate of crystalline compounds integrated into HPβCD was determined by the structure’s type and extent, as well as the drug’s physicochemical properties [40]. The solubility profile of VRC was greatly influenced by its interactions with HPβCD. Therefore, X-ray diffraction may be utilized to analyze complexation by comparing changes in the different peaks of the complex to those of pure drug molecules. X-ray thermal analysis concepts are used to understand the XRD complexation mechanism, evaluating the disappearance or change of unique peaks. Drug amorphization is a crucial strategy for improving pharmaceutical water solubility. The distinctive high-intensity VRC peaks were superimposed or decreased in an amorphous profile of HPβCD polymeric nanobeads, indicating a loss of crystallinity [41,42].

Drug release from polymeric nanobeads was rapid and was influenced by the monomer, polymer, and pH of the media. The abrupt and maximum release was seen within 3 h (CDN5), whereas the overall range was 11.8–91%. Maximum and rapid release profiles were observed by increasing AMPS concentration (3–7%) in CDN5. This effect was not the same as a further increase in AMPS concentration (7–8%); VRC release was decreased from 8.4–84.7% (CDN7) because it suppresses release due to absorptivity of media. Conversely, there was a slight fall in VRC release (8.7–80.4%) by increasing MBA concentration (0.6–0.9%) in CDN9. It inoculates stiffness to the cross-linked structure of polymeric nanobeads, which decreases viable gaps between chains. Similarly, the rapid and abrupt release was observed at pH 6.8 due to the pH-independent swelling behavior of AMPS. Slightly more release was observed at pH 6.8 as compared to pH 1.2. Increasing the concentration of AMPS, MBA, and HPβCD at pH 6.8 exerts the same effect as at pH 1.2. K. Hosseinzadeh and H Sadeghi synthesized superabsorbent hydrogel based on starch and evaluated its swelling and release pattern. Our findings of swelling and release behaviour are relative to their findings, i.e., increasing AMPS content up to the limit, increased swelling and drug release, and Increasing MBA concentration decrease the drug dissolution of VRC [43]. However, the pure VRC released was slow at pH 1.2 and 6.8 due to a lack of solubility in the polymeric network. Drug release kinetics showed that at pH 1.2 and pH 6.8, VRC HPβCD based polymeric nanobeads were seen first order as the best-fit model indicated by R^2^ values (0.99). Furthermore, at pH 1.2, the value of “n” was 0.67, and at pH 6.8, it was 0.68, indicating non-Fickian diffusion. The small particle size, increased wettability, large porous surface area, and pH-independent swelling behavior all affected the dissolution rate [29].

The HPβCD based polymeric gel has been demonstrated to have no tissue-irritating effects following oral administration and enhanced solubility of the hydrophobic drug. The plausible binding mode of the VRC with the HPβCD and AMPS molecules. Molecular docking studies predicted the nanogel formation agent had wrapped the VRC, masking the electronegative chloride ions, which explains the sharp changes in the angles observed in XRD. Further, the hydrophobic side chains of the AMPS and HPβCD are involved in hydrophobic interaction with the azoles and difluoro-benzene rings and amalgamation of the compounds highlighting the hydrophobic patching mediated by the nanogel formulation. The MIC and MFC of CDN5 against *Candida albicans* were 0.82 µg/mL and 0.34 µg/mL, while *Aspergillus flavus* were 1.6 and 3.2. Similarly, for *Aspergillus fumigatus,* the MIC and MFC significantly decreased compared to pure VRC MIC and MFC.

## 5. Conclusions

VRC HPβCD based polymeric nanobeads were successfully prepared with pH-independent swelling features. Nanobeads were transparent and had excellent mechanical strength. The solubility of VRC was effectively enhanced in water and at pH 1.2 and 6.8. Drug release kinetic studies showed first order as the best-fit model predicted on R^2^ values with a non-Fickian diffusion mechanism of release. So, therefore, based on the results, it can be concluded that HPβCD polymeric nanobeads can be used as another potential, non-toxic and promising tool in addition to traditional complex systems.

## Figures and Tables

**Figure 1 pharmaceutics-15-00389-f001:**
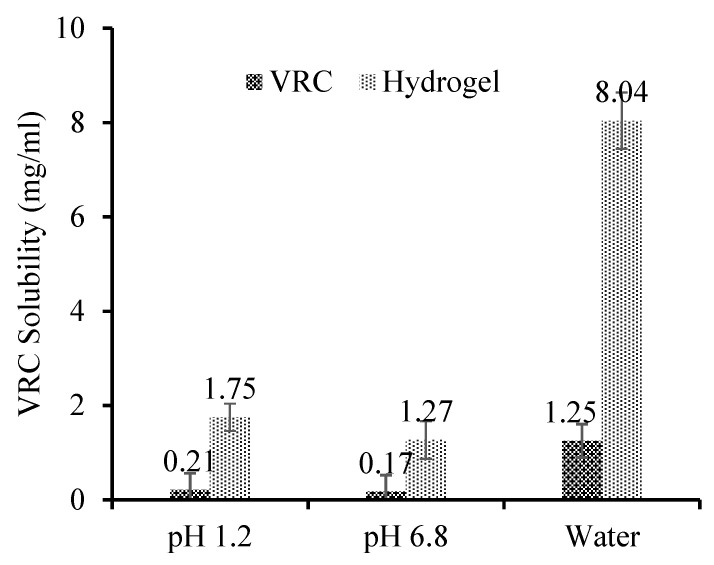
Solubility studies at pH 1.2, pH 7,4, and in water (CDN5) (Mean ± SD, *n* = 5).

**Figure 2 pharmaceutics-15-00389-f002:**
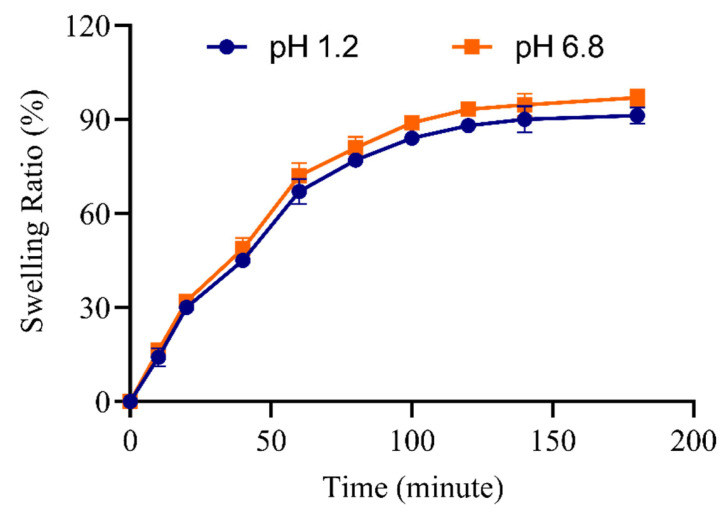
pH effect on % equilibrium swelling (CDN5) (Mean ± SD, *n* = 5).

**Figure 3 pharmaceutics-15-00389-f003:**
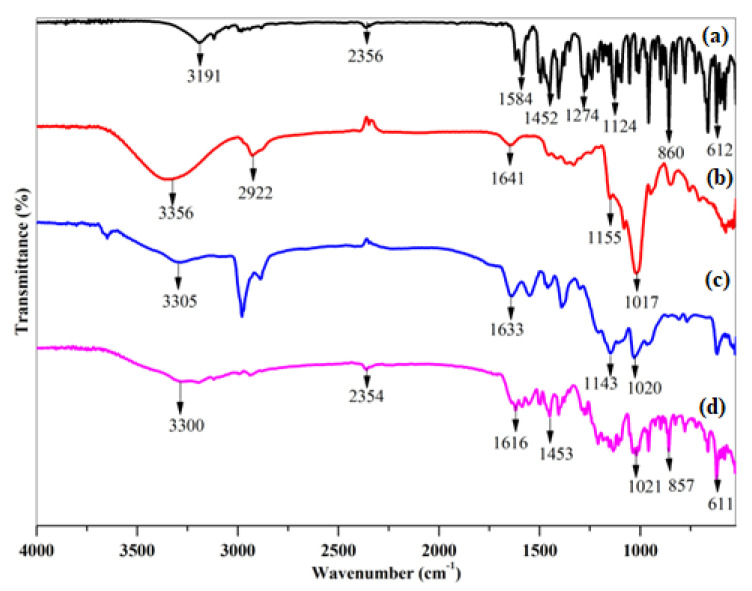
FTIR Spectra of pure VRC (**a**), HPβCD (**b**) and Blank (**c**) and loaded polymeric nanobeads (**d**).

**Figure 4 pharmaceutics-15-00389-f004:**
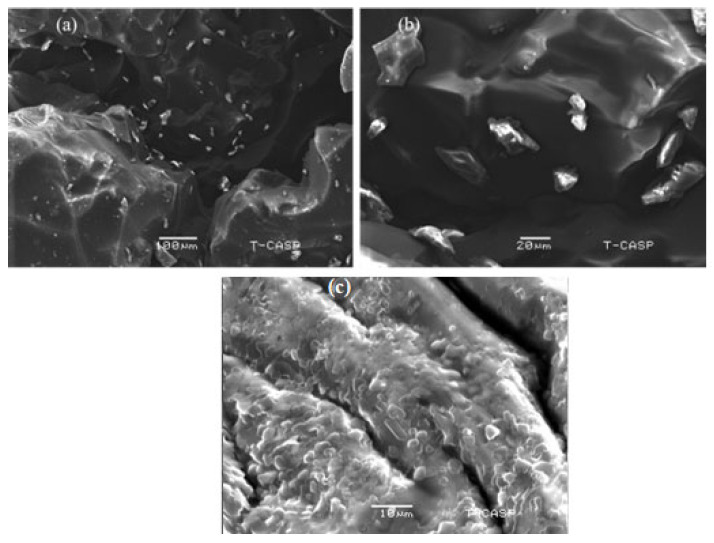
SEM of drug loaded nanogel at 100 µm (**a**), 20 µm (**b**), and blank nanogel at 10 µm (**c**) formulations.

**Figure 5 pharmaceutics-15-00389-f005:**
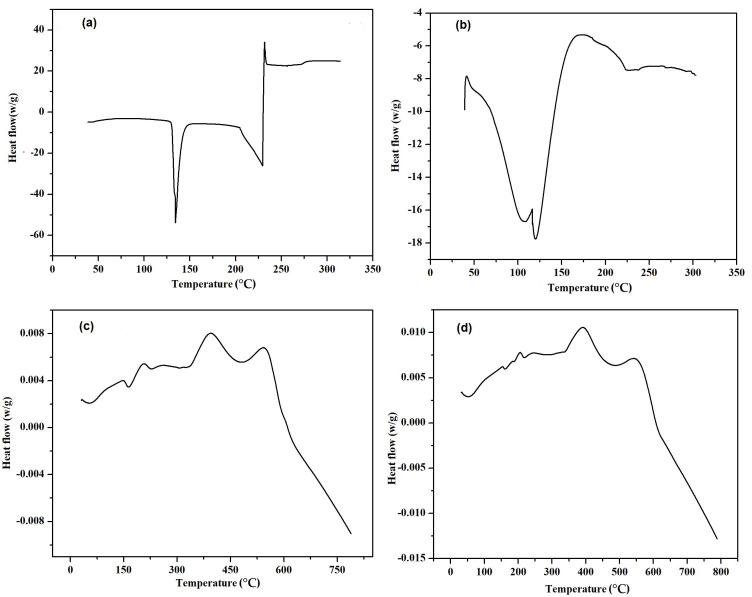
DSC of VRC (**a**), HPβCD (**b**), Blank (**c**), and loaded (**d**) HPβCD based polymeric nanobeads.

**Figure 6 pharmaceutics-15-00389-f006:**
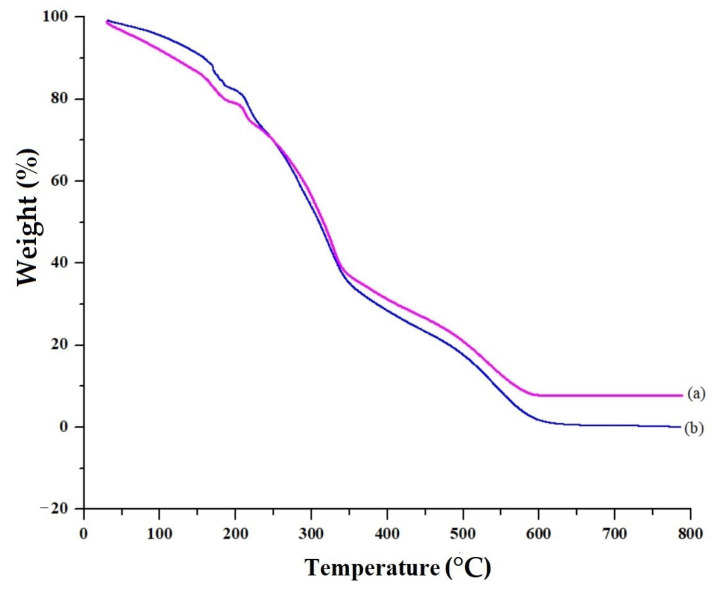
TGA of Blank (**a**) and loaded (**b**) HPβCD based polymeric nanobeads.

**Figure 7 pharmaceutics-15-00389-f007:**
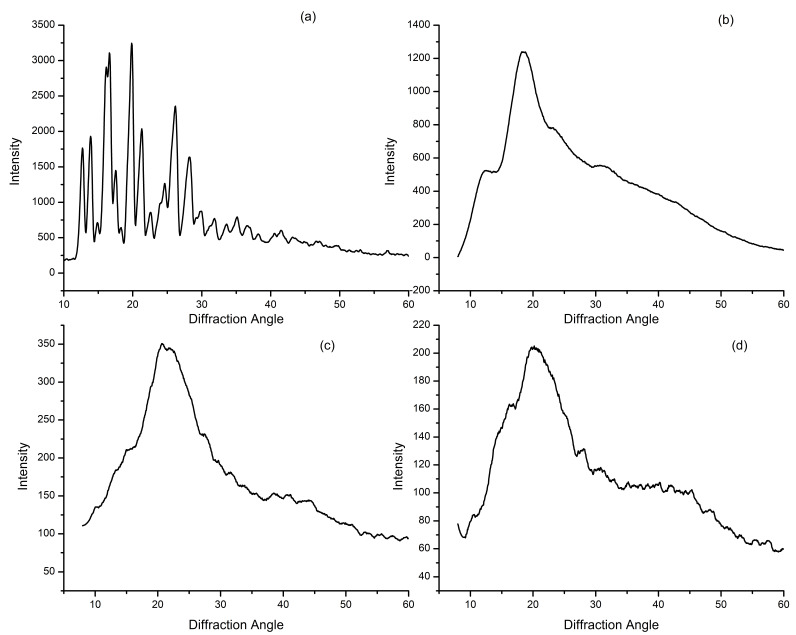
XRD diffractogram of VRC (**a**), HPβCD (**b**), Blank (**c**), and Loaded HPβCD based polymeric nanobeads (**d**).

**Figure 8 pharmaceutics-15-00389-f008:**
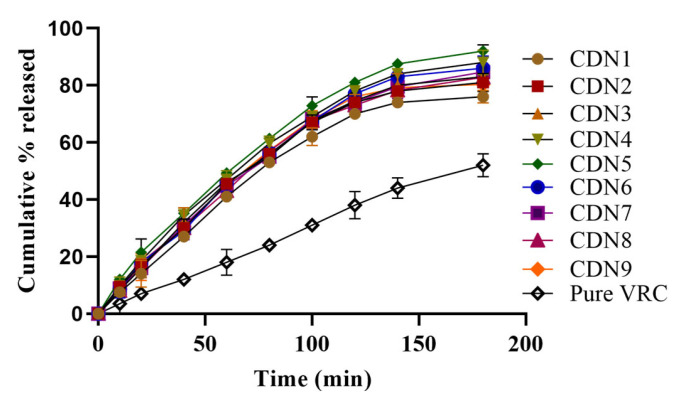
Effect of pH 1.2 on VRC release.

**Figure 9 pharmaceutics-15-00389-f009:**
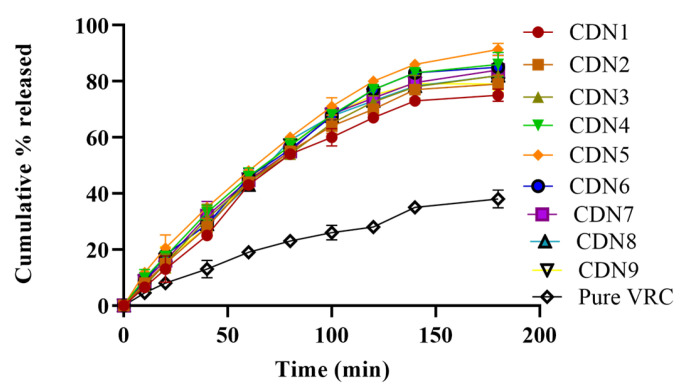
Effect of pH 6.8 on VRC release.

**Figure 10 pharmaceutics-15-00389-f010:**
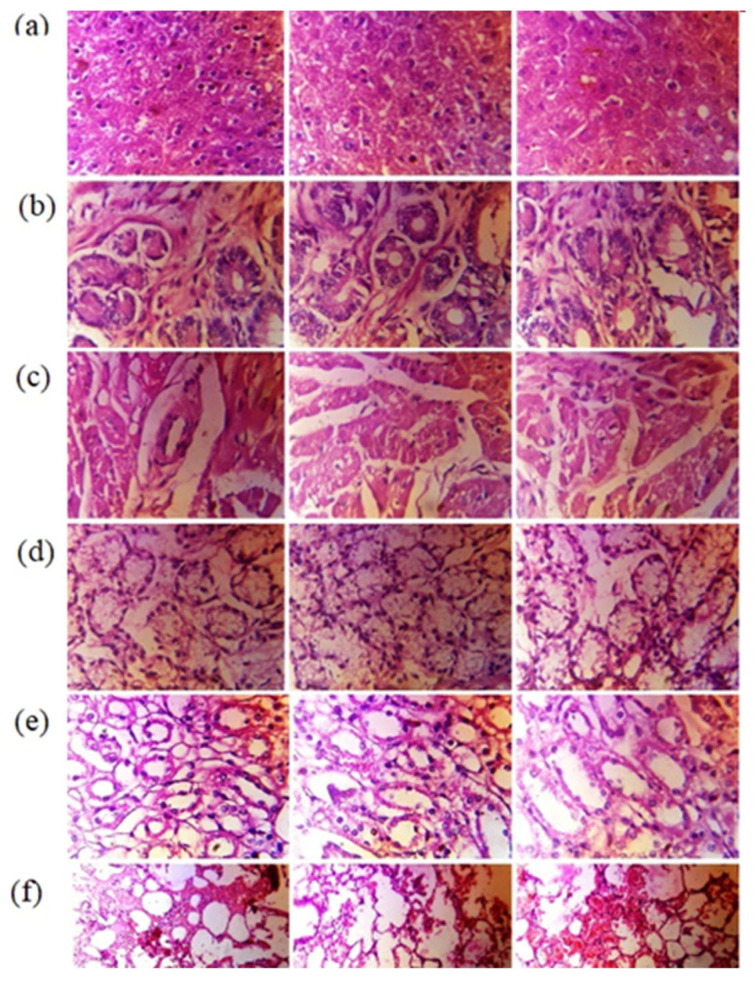
Histopathological examination of rabbit organs of Liver (**a**), stomach (**b**) heart (**c**), large intestine (**d**), kidney (**e**) and lungs (**f**).

**Figure 11 pharmaceutics-15-00389-f011:**
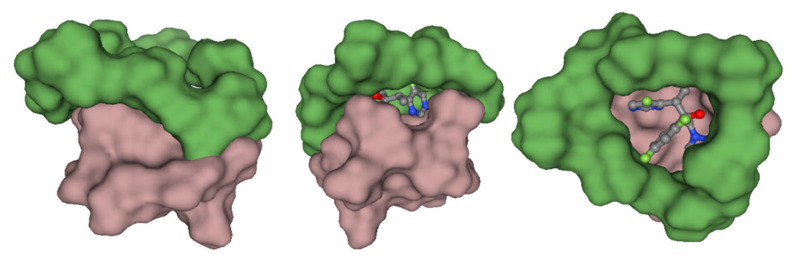
The CDN5 complex as obtained from molecular modelling. The VRC is presented as balls and sticks, while AMPS and HPβCD are shown as molecular surfaces. The visual was rendered using NGL viewer.

**Figure 12 pharmaceutics-15-00389-f012:**
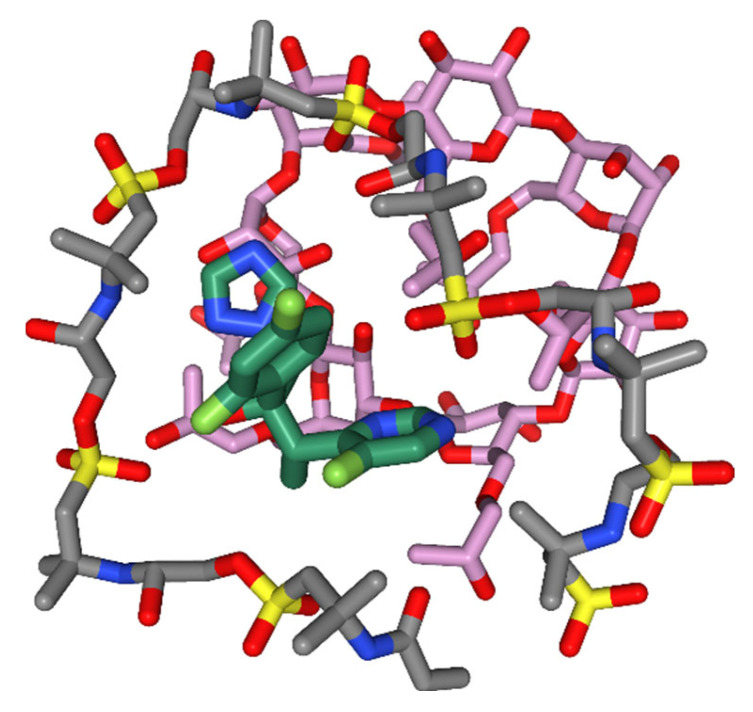
The amalgamation of CDN5 showing the hydrophobic contacts between the molecules.

**Figure 13 pharmaceutics-15-00389-f013:**
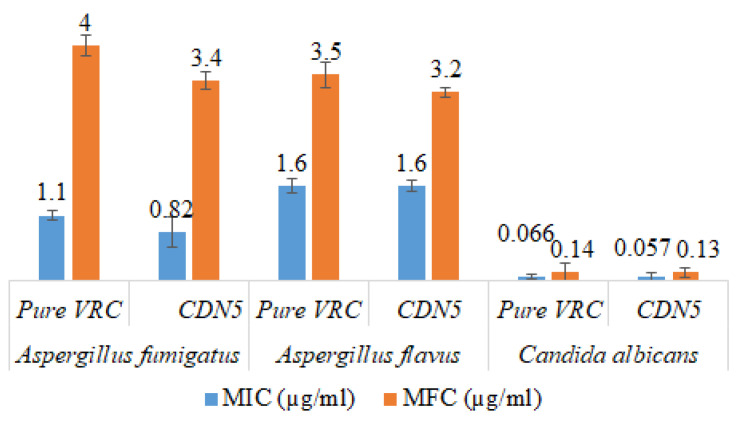
Antifungal activity of CDN5 and pure VRC against various fungal strains (Mean ± SD, *n* = 5).

**Table 1 pharmaceutics-15-00389-t001:** Composition of CD-based polymeric nanobeads.

Sample No.	HPβCD % *w/v*	AMPS % *w*/*v*	MBA % *w*/*v*	APS % *w*/*v*
CDN1	0.5	3	0.5	1
CDN2	1	3	0.5	1
CDN3	2	3	0.5	1
CDN4	1	6	0.6	1
CDN5	1	7	0.6	1
CDN6	1	8	0.6	1
CDN7	1	6	0.7	1
CDN8	1	6	0.8	1
CDN9	1	6	0.9	1

**Table 2 pharmaceutics-15-00389-t002:** % EE and Product yield of polymeric nanobeads (Mean ± SD, *n* = 5).

Formulation Code	% EE	% Product Yield
CDN1	72 ± 1.05	88 ± 1.05
CDN2	76 ± 0.25	93 ± 0.25
CDN3	81 ± 0.16	91 ± 0.16
CDN4	79 ± 0.23	92 ± 0.23
CDN5	87 ± 0.15	94 ± 0.15
CDN6	81 ± 0.11	85 ± 0.03
CDN7	80 ± 0.05	88 ± 0.17
CDN8	79 ± 0.15	83 ± 0.10
CDN9	80 ± 0.61	82 ± 0.61

**Table 3 pharmaceutics-15-00389-t003:** Particle size, zeta potential, and PDI of nanovesicles (Mean ± SD, *n* = 5).

Sample Code	Zeta Potential (mV)	Particles Size (nm)	PDI
CDN1	−31.0 ± 0.21	233.9 ± 010	0.21 ± 0.04
CDN2	−26.2 ± 0.31	329.8 ± 0.06	0.35 ± 0.11
CDN3	−22.0 ± 0.14	383.4 ± 0.11	0.36 ± 0.04
CDN4	−27.4 ± 0.50	335.5 ± 0.03	0.25 ± 0.09
CDN5	−35.1 ± 0.13	220.2 ± 0.06	0.23 ± 0.02
CDN6	−26.5 ± 0.21	400.2 ± 0.13	0.35 ± 0.05
CDN7	−29.3 ± 0.32	436.3 ± 0.10	0.26 ± 0.02
CDN8	−30.3 ± 012	442.5 ± 0.11	0.29 ± 0.06
CDN9	−29.4 ± 0.09	390.1 ± 0.50	0.24 ± 0.14

**Table 4 pharmaceutics-15-00389-t004:** Drug release kinetics results of CD nanobeads and pure VRC at pH 1.2 and pH 6.8.

Kinetics Model		pH 1.2		pH 6.8	
	Parameters	CDN1 to CDN9 (Mean)	Pure VRC	CDN1 to CDN9 (Mean)	Pure VRC
Zero Order	R^2^	0.98	0.99	0.89	0.97
	T_50_	94.6	164	96	266
	T_90_	170	296	173	300
First Order	R^2^	0.99	0.98	0.99	0.99
	T_50_	76.0	179	78.2	173
	T_90_	252	597	219	575
Higuchi	R^2^	0.93	0.87	0.93	0.88
Korsmeyer-Peppas	R^2^	0.97	0.99	0.96	0.98
	n	0.67	0.95	0.68	0.88

**Table 5 pharmaceutics-15-00389-t005:** Hematological Analysis of rabbit (Mean ± SD, *n* = 5).

Parameters	Group 1	Group 2	Group 3
Haemoglobin (g/dL)	10.1 ± 1.30	12.2 ± 0.88	10.9 ± 1.66
RBCs count (10^6^/µL)	4.9 ± 0.86	6.4 ± 1.73	5.6 ± 1.79
WBCs count (10^3^/µL)	9.7 ± 1.50	8.3 ± 1.16	8.7 ± 1.18
Platelets (10^3^/µL)	282 ± 3.40	230 ± 2.62	236 ± 1.97
PCV/HCT (%)	32.6 ± 1.32	39.2 ± 0.97	34 ± 0.84
MCV (fl)	66.9 ± 4.14	61.3 ± 5.51	64.1 ± 3.90
MCH (pg)	20.7 ± 0.23	19.1 ± 0.84	19.6 ± 1.32
MCHC (g/dL)	31.0 ± 1.30	31.1 ± 2.32	31.3 ± 1.88
Neutrophils (%)	44.4 ± 1.61	47.8 ± 1.61	46.8 ± 1.73
Lymphocytes (%)	41.6 ±0.76	44.8 ± 0.55	43.7 ± 0.78
Monocytes (%)	12.5 ± 1.70	10.5 ± 1.49	11.8 ± 1.51
Eosinophils (%)	1.2 ± 0.53	0.9 ± 0.67	0.7 ± 0.81

**Table 6 pharmaceutics-15-00389-t006:** Weight of organs from necropsied rabbits of all groups. (Mean ± SD, *n* = 5).

Groups Name	Heart (g)	Kidney (g)	Liver (g)	Lungs (g)	Stomach (g)	Small Intestine (g)
Group A	3.5 ± 0.05	7.5 ± 0.03	50 ± 1.02	11.5 ± 0.14	25 ± 0.20	14 ± 2.10
Group B	3.7 ± 0.01	7.4 ± 0.16	49.6 ± 1.5	11.3 ± 1.70	24.7 ± 0.08	13.5 ± 1.11
Group C	3.3 ± 0.11	7.9 ± 0.31	48.0 ± 0.09	10.9 ± 1.12	24.3 ± 0.11	14.6 ± 0.09

## Data Availability

The data presented in this study are available from the authors upon reasonable request.

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
