# Peer review of "Voriconazole Cyclodextrin Based Polymeric Nanobeads for Enhanced Solubility and Activity: In Vitro/In Vivo and Molecular Simulation Approach"

_pharmaceutics, 2023, doi:10.3390/pharmaceutics15020389_

Round 1

Reviewer 1 Report

The research work on “Voriconazole cyclodextrin based polymeric nanobeads for enhanced solubility and activity: In-vitro/In-Vivo and 3 Molecular Simulation approach” is in the scope of the journal. In this work Hydroxypropyl β-cyclodextrin (HPβCD) based polymeric nanobeads containing voriconazole (VRC) were fabricated by free radical polymerization using N, N′-methylene bisacrylamide (MBA) as a crosslinker, 2-acrylamide-2-methylpropane sulfonic acid (AMPS) as monomer and ammonium persulfate (APS) as reaction promoter. The team of researchers concluded that developing polymeric nanobeads can be a promising tool to enhance the solubility and efficacy of hydrophobic drugs like VRC besides decreased toxicity and effective management of fungal infections.

Please find the observations and comments as follows:

1.       Is 2-acrylamide-2-methyl- 84 propane sulfonic acid (AMPS) safe to use for drug delivery? If yes up to what concentration and time?

2.       Cite all Characterization methods with an appropriate method like done in 2.3.1, if the method used is not novel or innovative.

3.       2.8. Antifungal activity, Why is PEG used as a solvent?

4.       3.1, Can the author validate the following sentences, with available literature or theory,

a.       Polymer concentrations affect the % EE; % EE increases with increasing polymer 258 concentration.

b.       The formulation CDN5 showed a maximum % EE due to the repulsion of 259 sulfonic acid groups.

5.       A is b and B is a in this figure. Figure 4: SEM of drug-loaded nanogel at 20 µm (a), 100 µm (b), and blank nanogel at 100 µm(c) 312 formulations.

6.       Figure 4: From SEM image beads (both at 20 and 100 microns) look bigger than the particle size in table 3. Please submit other images with more clarity if available.

7.       Please mentions DSC, TGA, and P XRD instrument details, make, and model.

8.       Figure 6: DSC of VRC (a), HPβCD (b), Blank (c), and loaded (d) HPβCD based polymeric gel., not found in images c and d.

9.       Validate/compare the results obtained with available results and cite at respective places in during discussion for better understanding to the readers.

Author Response

Response to Reviewer 1 Comments

Point 1: Is 2-acrylamide-2-methyl- 84 propane sulfonic acid (AMPS) safe to use for drug delivery? If yes up to what concentration and time?

Thanks for highlighting the point. Yes, AMPS is safe for oral delivery. Its LD50 for oral dosage forms is 1830mg/kg, whereas, for dermal products, it is 2000mg/kg. These values are for higher than amount used in one study. So its usage is quite safe in the reported research. Various references cited from the literature are attached here. It has been reported to have high hydrophilicity and biocompatibility.

  • Khalid, Q., Ahmad, M., & Usman Minhas, M. (2018). Hydroxypropyl‐β‐cyclodextrin hybrid nanogels as nano‐drug delivery carriers to enhance the solubility of dexibuprofen: Characterization, in vitro release, and acute oral toxicity studies. Advances in Polymer Technology37(6), 2171-2185.

Point 2: Cite all Characterization methods with an appropriate method like done in 2.3.1, if the method used is not novel or innovative.

Thank you for your suggestion. All characterization methods are cited in the text.

Point 3: 2.8. Antifungal activity, why is PEG used as a solvent?

PEG has already been used as a solvent for performing the antifungal activity of the hydrophobic drugs. The reference is cited here.

  • Ostrosky-Zeichner, L., Rex, J. H., Pappas, P. G., Hamill, R. J., Larsen, R. A., Horowitz, H. W., & Lee, J. (2003). Antifungal susceptibility survey of 2,000 bloodstream Candida isolates in the United States. Antimicrobial agents and chemotherapy47(10), 3149-3154.

Point 4: 3.1, Can the author validate the following sentences, with available literature or theory,

  1. Polymer concentrations affect the % EE; % EE increases with increasing polymer 258 concentration.
  2. The formulation CDN5 showed a maximum % EE due to the repulsion of 259 sulfonic acid groups.

Thank you for valuable comments. The data for aforesaid results is cited from available literature and is already added in the manuscript.

  • Rizvi, S. S. B., Akhtar, N., Minhas, M. U., Mahmood, A., & Khan, K. U. (2022). Synthesis and Characterization of Carboxymethyl Chitosan Nanosponges with Cyclodextrin Blends for Drug Solubility Improvement. Gels8(1), 55.
  • Sarfraz, R. M., Ahmad, M., Mahmood, A., & Ijaz, H. (2018). Development, in vitro and in vivo evaluation of pH responsive β-CD-comethacrylic acid-crosslinked polymeric microparticulate system for solubility enhancement of rosuvastatin calcium. Polymer-Plastics Technology and Engineering57(12), 1175-1187.

Point 5: A is b and B is a in this figure. Figure 4: SEM of drug-loaded nanogel at 20 µm (a), 100 µm (b), and blank nanogel at 100 µm(c) 312 formulations.

Figures are corrected as suggested.               

Point 6: Figure 4: From SEM image beads (both at 20 and 100 microns) look bigger than the particle size in table 3. Please submit other images with more clarity if available.

Thank you for your suggestion. The size obtained is based on the resolutions from the microscope. Image with more clarity is added in the manuscript as suggested.

Point 7: Please mention DSC, TGA, and P XRD instrument details, make, and model.

The models for the aforesaid instruments are mentioned in the manuscript file as suggested,

Point 8: Figure 6: DSC of VRC (a), HPβCD (b), Blank (c), and loaded (d) HPβCD based polymeric gel. not found in images c and d.

Figure captions are corrected as advised.

Point 8: Validate/compare the results obtained with available results and cite at respective places in during discussion for better understanding to the readers.

As suggested, citations have been added and placed at relevant places.

Reviewer 2 Report

The manuscript submitted must be improved before publication. Many significant points need the authors' attention. A few comments are presented below; addressing them will improve the quality of the Manuscript.

1. Remove AMPS and APS from keywords or include full chemical name in keywords

2. Rewrite lines 48-49

3. Line 86, is swelling of polymer pH-dependent or pH-independent?

4. line 109 - Was the reaction carried out at 450 °C?

5. line 113 - Was drying done at 450 °C - 650 °C?

6. No Drying step after washing of polymer in line 116?

7. Line 127, please use uniform terminology: nanobeads or Nanogel beads.

8. Represent the chemical structure of the proposed reaction; it is easy o visualize the chemical reaction if represented properly.

9. Section 2.3.1- Orgaing solvent, in which the drug has excellent solubility, is always used for % EE, as it can easily extract all the drugs from the complex. If water or buffer is used, like in this case, there are chances that all drugs will not be extracted and % EE results will be wrong.

10. Provide UV method development as it is the primary method for drug estimation.

11. PSD and zeta potential methods provide sample preparation methods.

12. line 177 - 30/min is what? And 50 - 600 ° is wrong. Please correct

13. Section 2.4 - pH 6.8 is not considered basic media

14. Section 2.6 - Way Rabbits were used for acute toxicity studies instead of rats and mice. Please provide proper justification and also provide ethical board approval for the use of rabbits for such a small study which can be easi done on rats and mice. Killing 15 rabbits for the acute study is not justifiable. 

15. section 2.8 - Why were VRC and polymer beads solubilized in PEG? which molecular weight was PEG used? At what concentration was the study performed? The method is not clear and needs extensive modification.

16. line 291 - There is no 1536 peak in any of the FTIR spectra shown in figure 3, and the formation of an ether bond, as mentioned in line 291, means there is covalent bonding between the drug and polymer.

17. Particle size of polymer nanobeads is 220-436 nm, as reported in PSD, but the SEM image is 20 and 100 µm. Which makes a particle of 200 nm seems invisible or very small.  Please include zoomed-in images of SEM or remove them as it does not impart any additional details in the manuscript.

18. Section 3.6 - SD of Particle size is very small. Please provide the graphical peaks to support these results.

19. Figure 6 - missing two DSC chromatograms in figures c and d.

20. Figure 6 - why is there no endothermic peak of VRC at 133 °C corresponding to its melting peak in the DSC?

21. Figure 9 - dissolution curve for pure VRC is missing from figure 8 and figure 9; it is important for comparison purposes.

22. Section 3.11 - lines 386-387, toxicity studies are not the way to check the unreacted monomer and initiator. HPLC analysis has to be performed on synthesized material to check that no toxic monomer or initiator is present before acute toxicity studies. 

23. Molecular docking study results are unclear; please include more data like the number of bonds forming between drug and polymer, type of bonds forming, length of bonds, free energy calculations, etc. Without this information, figure 11 and figure 12 are not adding any useful information about the complex formation. 

24. Please represent Figure 13 properly.

25. Statistical calculation is required for to compare the MIC and MFC of VRC and nanobead VRC. There are no standard deviations in the figures. The value 0.34 µg/mL mentioned in line 461 is not seen anywhere in figure 13.

26. line 483, the solubility of CDN5 is drastically increased in water compared to pH 6.8 buffer; is there any specific reason for this? 

27. which buffers are used for pH 1.2 and pH 6.8 buffers? It is not mentioned anywhere in the manuscript.

28. from figure 9. it seems there is no statistical difference in the release of VRC from any of the CDNs. Please provide statistical analysis to justify the dissolution in the discussion section.

Author Response

Response to Reviewer 1 Comments

Point 1: Remove AMPS and APS from keywords or include full chemical name in keywords

As suggested, the changes have been made in the abstract.

Point 2: Rewrite lines 48-49

The lines are rewritten as suggested.

Point 3: Line 86, is swelling of polymer pH-dependent or pH-independent?

Thank you for kind suggestion. Swelling of complex formation can be pH dependent as well as pH independent. Nature of monomer or polymer plays key role in deciding the dependence on pH. In case of AMPS, swelling is pH independent. Swelling of AMPS in complex formation is highly dependent on ionizable sulfonate groups. If Polymethacrylic acid is used, its swelling depends on the ionizable functional groups present in formulation therefore complex formation will be pH dependent.

  • Asghar, S., Akhtar, N., Minhas, M. U., & Khan, K. U. (2021). Bi-polymeric spongy matrices through cross-linking polymerization: synthesized and evaluated for solubility enhancement of acyclovir. AAPS PharmSciTech22(5), 1-16.
  • Sarfraz, R. M., Khan, M. U., Mahmood, A., Akram, M. R., Minhas, M. U., Qaisar, M. N., & Zaman, M. (2020). Synthesis of co-polymeric network of carbopol-g-methacrylic acid nanogels drug carrier system for gastro-protective delivery of ketoprofen and its evaluation. Polymer-Plastics Technology and Materials59(10), 1109-1123.

Point 4: line 109 - Was the reaction carried out at 450 °C?

Correction is done in the manuscript file for temperature.

Point 5: Line 113 - Was drying done at 450 °C - 650 °C?

Correction for drying temperature is done in the manuscript file.

Point 6: No Drying step after washing polymer in line 116?

Drying was done using lyophilization and is already mentioned in the manuscript.

Point 7: Line 127, please use uniform terminology: nanobeads or Nanogel beads.

The terminology is corrected as suggested.

Point 8: Represent the chemical structure of the proposed reaction; it is easy to visualize the chemical reaction if represented properly.

Chemical structure of the proposed reaction is added as advised in supplementary file.

Point 9:  Section 2.3.1- Orgaing solvent, in which the drug has excellent solubility, is always used for % EE, as it can easily extract all the drugs from the complex. If water or buffer is used, like in this case, there are chances that all drugs will not be extracted and % EE results will be wrong.

Thank you for your valuable suggestion. In entrapment efficiency, ethanol and deionized water was used in 1:1 ratio. Same has been corrected in the manuscript file.

Point 10: Provide UV method development as it is the primary method for drug estimation.

UV/VIS spectroscopic method was validated using our spectrophotmeter following ICH guidelines and is cited here.

Adams, A. I., Steppe, M., Frehlich, P. E., & Bergold, A. M. (2006). Comparison of microbiological and UV-spectrophotometric assays for determination of voriconazole in tablets. Journal of AOAC International89(4), 960-965.

Point 11. PSD and zeta potential methods provide sample preparation methods.

Sample preparation method has been added in the manuscript file as advised,

Point 12: line 177 - 30/min is what? And 50 - 600 ° is wrong. Please correct

The Aforesaid corrections are done in the manuscript file as suggested.

Point 13: Section 2.4 - pH 6.8 is not considered basic media.

      Thanks for pointing out. Corrections are done in the manuscript file.

Point 14: Section 2.6 - Way Rabbits were used for acute toxicity studies instead of rats and mice. Please provide proper justification and also provide ethical board approval for the use of rabbits for such a small study which can be easily done on rats and mice. Killing 15 rabbits for the acute study is not justifiable. 

            Rabbits were used for studies owing to easy availability in the animal house of faculty of Pharmacy. Due to breeding of rats and mice, they were not available in sufficient number       so rabbits were used for the study and same was approved also by the Ethical Committee           of Pharmacy as well as Bahauddin Zakariya University, Multan. Ethical Board approval is           attached with the manuscript file already.

Point 15: Section 2.8 - Why were VRC and polymer beads solubilized in PEG? Which molecular weight was PEG used? At what concentration was the study performed? The method is not clear and needs extensive modification.

VRC was solubilized in PEG owing to already published data in the reference cited here. PEG used was of MW1450 whereas concentration used was 0.5 %.

  • Ostrosky-Zeichner, L., Rex, J. H., Pappas, P. G., Hamill, R. J., Larsen, R. A., Horowitz, H. W., & Lee, J. (2003). Antifungal susceptibility survey of 2,000 bloodstream Candida isolates in the United States. Antimicrobial agents and chemotherapy47(10), 3149-3154.

Point 16:. line 291 - There is no 1536 peak in any of the FTIR spectra shown in figure 3, and the formation of an ether bond, as mentioned in line 291, means there is covalent bonding between the drug and polymer.

As suggested, the corrections are done in the manuscript file.

Point 17: Particle size of polymer nanobeads is 220-436 nm, as reported in PSD, but the SEM image is 20 and 100 µm. Which makes a particle of 200 nm seems invisible or very small.  Please include zoomed-in images of SEM or remove them as it does not impart any additional details in the manuscript.

The image has been replaced with better resolution images as advised by reviewer.

Point 18: Section 3.6 - SD of Particle size is very small. Please provide the graphical peaks to support these results.

Graphical peaks are added in the manuscript as advised.

Point 19: Figure 6 - missing two DSC chromatograms in figures c and d.

DSC chromatograms are added in the Figure 6 as pointed out by reviewer.

Point 20. Figure 6 - why is there no endothermic peak of VRC at 133 °C corresponding to its melting peak in the DSC?

DSC explanation has been added according to the Figure 5 and all the corrections including Figure captions are incorporated in the manuscript file.

  1. Figure 9 - dissolution curve for pure VRC is missing from figure 8 and figure 9; it is important for comparison purposes.

Dissolution curve for pure VRC is added in the Figures ad advised.

Point 22. Section 3.11 - lines 386-387, toxicity studies are not the way to check the unreacted monomer and initiator. HPLC analysis has to be performed on synthesized material to check that no toxic monomer or initiator is present before acute toxicity studies. 

Toxicity parameters were performed on the basis of literature as cited in the manuscript and acute cytotoxic studies were performed and various hepatic and renal parameters were also performed. These are added in the revised manuscript file.  

Point 23. Molecular docking study results are unclear; please include more data like the number of bonds forming between drug and polymer, type of bonds forming, length of bonds, free energy calculations, etc. Without this information, figure 11 and figure 12 are not adding any useful information about the complex formation. 

Hydrophobic interaction between the aliphatic chains mostly stabilize the complex and there is no standard hydrogen bond in between the drug and polymer. The docking simulation section has been revised in the manuscript file to reflect this information.

Point 24. Please represent Figure 13 properly.

Figure is properly represented as suggested.

Point 25. Statistical calculation is required for to compare the MIC and MFC of VRC and nanobead VRC. There are no standard deviations in the figures. The value 0.34 µg/mL mentioned in line 461 is not seen anywhere in figure 13.

Corrections are done as advised by Reviewer.

Point 26. line 483, the solubility of CDN5 is drastically increased in water compared to pH 6.8 buffer; is there any specific reason for this? 

Complexation using CD is major reason for increased solubility. CDs have been reported to enhance solubility of many compounds following complexation. Reference is also cited here.  

Khalid, Q., Ahmad, M., & Usman Minhas, M. (2018). Hydroxypropyl‐β‐cyclodextrin hybrid nanogels as nano‐drug delivery carriers to enhance the solubility of dexibuprofen: Characterization, in vitro release, and acute oral toxicity studies. Advances in Polymer Technology37(6), 2171-2185.

Point 27. Which buffers are used for pH 1.2 and pH 6.8 buffers? It is not mentioned anywhere in the manuscript.

pH 1.2 buffer used was HCl buffer whereas pH 6.8 was phosphate buffer.

Point 28. from figure 9. it seems there is no statistical difference in the release of VRC from any of the CDNs. Please provide statistical analysis to justify the dissolution in the discussion section.

Statistical analysis is added in the discussion section for the release profile of VRC from CDNs.

Reviewer 3 Report

The manuscript reports a detailed study on the encapsulation of the hydrophobic drug Voriconazole (VRC), in polymeric nanobeads hydroxypropyl with incorporated β-cyclodextrin (HPβCD) for system design purposes for enhanced solubility and bioactivity.

The work presented here is interesting and the experiments have been performed correctly. The manuscript is very well written and easy to read. It is necessary to emphasize that the cited literature is very well processed and used in an exemplary way in explaining the experimental result.

I think this manuscript is well written and showed clear results with detailed characterization of the system for the purpose of storage and controlled storage and drug realization. Furthermore, are described In-vitro/In-Vivo study analysis and the study of molecular dynamics.

In conclusion, this manuscript could be accepted for publication in the Pharmaceutics journal at the present stage.

Author Response

The manuscript reports a detailed study on the encapsulation of the hydrophobic drug Voriconazole (VRC), in polymeric nanobeads hydroxypropyl with incorporated β-cyclodextrin (HPβCD) for system design purposes for enhanced solubility and bioactivity.

The work presented here is interesting and the experiments have been performed correctly. The manuscript is very well written and easy to read. It is necessary to emphasize that the cited literature is very well processed and used in an exemplary way in explaining the experimental result.

I think this manuscript is well written and showed clear results with detailed characterization of the system for the purpose of storage and controlled storage and drug realization. Furthermore, are described In-vitro/In-Vivo study analysis and the study of molecular dynamics.

In conclusion, this manuscript could be accepted for publication in the Pharmaceutics journal at the present stage.

Round 2

Reviewer 2 Report

Point 9:  Section 2.3.1- Orgaing solvent, in which the drug has excellent solubility, is always used for % EE, as it can easily extract all the drugs from the complex. If water or buffer is used, like in this case, there are chances that all drugs will not be extracted and % EE results will be wrong.

Thank you for your valuable suggestion. In entrapment efficiency, ethanol and deionized water was used in 1:1 ratio. The same has been corrected in the manuscript file.

Were the experiments performed again? Please perform the experiment again before changing the method in manuscript.

Point 11. PSD and zeta potential methods provide sample preparation methods.

Sample preparation method has been added in the manuscript file as advised,

Need sample preparation method such as dilution and what conc of sample used for PSD and zeta analysis.

Point 14: Section 2.6 - Way Rabbits were used for acute toxicity studies instead of rats and mice. Please provide proper justification and also provide ethical board approval for the use of rabbits for such a small study which can be easily done on rats and mice. Killing 15 rabbits for the acute study is not justifiable. 

            Rabbits were used for studies owing to easy availability in the animal house of faculty of Pharmacy. Due to breeding of rats and mice, they were not available in sufficient number       so rabbits were used for the study and same was approved also by the Ethical Committee           of Pharmacy as well as Bahauddin Zakariya University, Multan. Ethical Board approval is           attached with the manuscript file already.

Availability of Rabbits generally is very difficult than Rats and Mice. However, availability should not be the reason for using higher animals than Rats and Mice for studies without testing in lower animals first.

Point 15: Section 2.8 - Why were VRC and polymer beads solubilized in PEG? Which molecular weight was PEG used? At what concentration was the study performed? The method is not clear and needs extensive modification.

VRC was solubilized in PEG owing to already published data in the reference cited here. PEG used was of MW1450 whereas concentration used was 0.5 %.

From the comment, I understand that a PEG solution was used for solubilization, which is unclear in the manuscript. The manuscript suggests use of pure PEG for the solubilization of the drug. Please make a clear sentence in the manuscript.